# REVERSIBLE LIFELONG MODEL EDITING VIA SEMANTIC ROUTING-BASED LORA

## ABSTRACT

Large Language Models (LLMs) have demonstrated remarkable capabilities in natural language processing. However, the dynamic evolution of real-world knowledge necessitates continual editing of specific knowledge within LLMs. While existing model editing methods explore modular isolation or parameter-efficient strategies, they often suffer from semantic drift or knowledge forgetting during sequential editing due to the continual updating of semantic content. To address these challenges, we propose **SoLA**, a **S**emantic r**o**uting-based **LA** framework for reversible lifelong model editing. In SoLA, each edit is encapsulated as an independent LoRA module which is frozen after training and a semantic routing record is established to map it to the input semantic representation, allowing dynamic activation of LoRA modules via semantic matching. This mechanism avoids semantic drift caused by clustering updating and mitigates catastrophic forgetting from parameter sharing. Importantly, SoLA supports both insertion and deletion of edits. By simply removing key from the semantic routing, specific edits can be precisely revoked, restoring the model's original behavior. To our knowledge, this reversible rollback editing capability is the first to be achieved in existing literature. Furthermore, SoLA integrates the decision-making process into the edited layer itself, eliminating the need for auxiliary routing networks and enabling end-to-end decision-making process. Extensive experiments across three representative tasks (document classification, question answering, and hallucination correction) demonstrate that SoLA effectively learns and retains edited knowledge, achieving accurate, efficient, and reversible lifelong model editing.

## 1 INTRODUCTION

Large language models (LLMs) have demonstrated remarkable capabilities in the field of natural language processing, attracting the attention of numerous researchersBrown et al. (2020)Achiam et al. (2023)Radford et al. (2019)Touvron et al. (2023). However, LLMs also face severe challenges, including hallucinationsHuang et al. (2025), biasesHartvigsen et al. (2022), and the generation of harmful contentDe Cao et al. (2021). Moreover, the dynamic nature of real-world knowledge requires the continuous updating of specific information in LLMsLazaridou et al. (2021). Re-training these models from scratch is both expensive and time-consumingBiderman et al. (2023). In this scenario, the demand for model editing is increasing with aims to modify specific knowledge in the training model continuously without re-training the model or affecting its performance on unrelated inputs, which is known as lifelong model editingYao et al. (2023)Hartvigsen et al. (2022)Yu et al. (2024).

The lifelong model editing aims to continuously update the model to adapt to the constantly changing world. However, most of the existing model editing methods are designed for single, isolated editingMitchell et al. (2021)Meng et al. (2022a). When applied to a continuous environment, these methods often lead to catastrophic forgetting of the previously learned knowledge in the LLMs, resulting in performance degradation and reduced model reliabilityYao et al. (2023)Gu et al. (2024). To solve this problem, recent methods propose freezing the original model parameters and introducing lightweight trainable modules to inject new knowledge. These methods usually retain most of the model's knowledge by keeping the base parameters fixed and combining the learnable modules for specific task updates. For example, MeloYu et al. (2024) determines the clustering centres of editors by building a neuron index and dynamically updates its semantic representation based on the

distance from the clustering centres. ELDERLi et al. (2025) adopts the Mixture-of-Experts (MoE) approach, assigning scores to each LoRA expert through a set of learnable LoRA allocation codes and selecting the top-k LoRA for calculation.

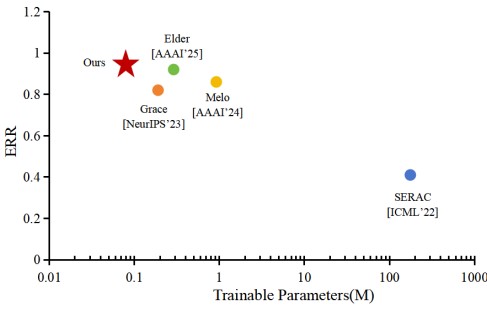 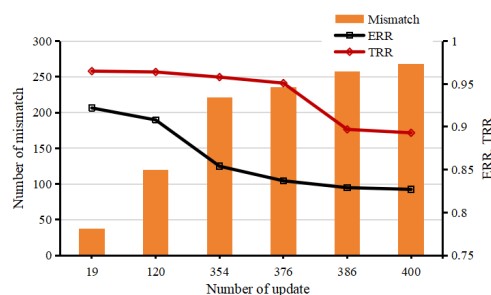

(a) Trainable parameters and ERR accuracy comparison.

(b) Number of mismatches of LoRA allocation and ERR/TRR in MELO.

Figure 1: Figure(a) presents the comparison of trainable parameters and ERR accuracy between SoLA (Ours) and other methods. Figure(b) indicates the number of mismatches in LoRA allocation with ERR/TRR accuracy in MELO. All experiments are conducted on Scotus datasets with the same backbone.

Although these methods show potential in improving parameter efficiency and reducing interference, they face inherent limitations when applied to lifelong model editing. Specifically, MELOYu et al. (2024) improves module separability through semantic clustering, but its clustering centers are updated during the editing process, which may lead to semantic drift and module matching errors. ELDERLi et al. (2025) employ MoE and selects the top-k routes to activate each editing subset, although this strategy has high parameter efficiency, shared and continuously updated parameters can lead to catastrophic forgetting - new editing may overwrite or interfere with existing editing.

To address these limitations, we propose **SoLA**, a **S**emantic r**o**uting-based **L**oR**A** framework for reversible lifelong model editing. In SoLA, we allocate an independent LoRA module for each edit and establish semantic routing to record the mapping relationship between LoRA module and the input semantic representation. During editing, the LoRA module fully learns for the current task and then remains frozen to retain the learned knowledge. The corresponding key is also not updated. During inference, the semantic representation of the input is calculated, and the corresponding LoRA module is dynamically activated through semantic routing. Since the LoRA module and the semantic representation have not been updated, **we fundamentally avoid catastrophic forgetting and semantic drift problems caused by continual updating**. At the same time, each edit only requires training for the current LoRA module, significantly reducing the computational resource consumption. As shown in Fig.1a, SoLA achieves optimal performance with only 0.08M additional parameters, significantly outperforming previous methods, highlighting SoLA's exceptional parameter efficiency and editing accuracy.

More importantly, by removing the semantic key from the mapping memory table, we can precisely revoke specific edits, allowing the model to recover its original behavior without re-training. This means we can freely add and delete edits, and truly **achieve controllable rollback and restoration of editing**. To our knowledge, this is the first time that controllable rollback of editing has been achieved in existing literature. Moreover, existing works usually require introducing auxiliary routing network outside the edited layer to determine whether to enable the LoRA module. In this paper, we propose a master decision-making mechanism that aggregates the decision-making process into the edited layer, avoiding the reliance on auxiliary routing network and achieving an end-to-end decision-making process.

Extensive experiments validate the effectiveness of our approach. Our main contributions are summarized as follows:

- We propose SoLA, a novel framework for reversible lifelong model editing, which incorporates a controllable LoRA mechanism guided by semantic routing. After each edit, both

the LoRA modules and the corresponding keys in the mapping memory are frozen, effectively mitigating catastrophic forgetting and semantic drift. Moreover, only the currently active LoRA modules are trained during editing, significantly reducing computational overhead.(Fig.1a).

- We employs semantic routing to establish a precise mapping between LoRA modules and semantic representations. By removing the associated key, any specific edit can be precisely revoked, enabling flexible addition and deletion of edits.

- we propose a master decision-making mechanism that aggregates the decision-making process into the edited layer, avoiding the reliance on an auxiliary routing network and achieving an end-to-end decision-making process.

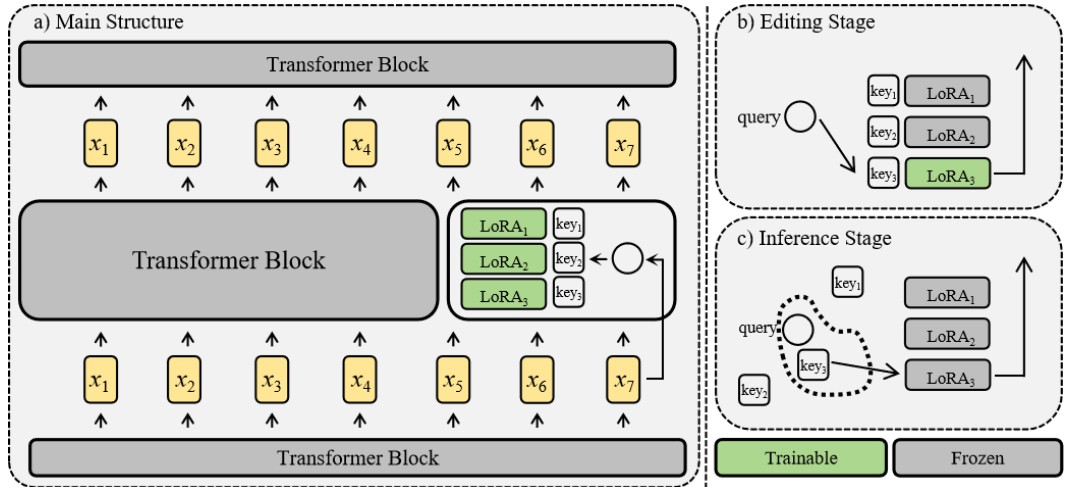

Figure 2: Main framework of our method. a) indicates the edit layer in model, where we retain the Transformer Block of base model frozen and add trainable LoRA module to the edited layer of base model. b) show the editing process, where every edit will be assigned LoRA module and the query of input will be mapped to assigned LoRA module. c) present the matching process between query of input and LoRA module in reference. Colour green indicates trainable, while color grey is frozen.

## 2 RELATED WORK

### 2.1 MODEL EDITING

Model Editing aims to update specific knowledge within LLMs while preserving their previous knowledge. Current model editing methods can be broadly categorized into three paradigms: Meta-Learning methods, Locate-then-Edit methods, and Memory-Based methods. Meta-Learning Methods employ a hypernetwork to predict the necessary gradients for editing and apply these gradients to the base model to achieve the updateMitchell et al. (2021). However, this approach typically requires additional data for training the hypernetwork. Locate-then-Edit methods first identify the neurons most critical for the current input via minor perturbations and then directly modify these neurons to update the model's knowledgeMeng et al. (2022a)Meng et al. (2022b). While effective, these methods are often cumbersome and exhibit limitations when handling a large volume of simultaneous knowledge updates. Memory-Based methods store the information associated with edits in an buffer, which acts as a patch model augmenting the original modelMitchell et al. (2022). A significant drawback is that each edit necessitates retraining, making it difficult to adapt to continuous editing scenarios, and the memory bank grows incrementally with each edit, leading to escalating storage overhead. Recent studies have also explored editing constraints to maintain model behavior stability. AlphaEditFang et al. (2024) introduces a null-space constrained approach. It aims to apply edits more precisely by constraining changes to the null space of irrelevant knowledge.

However, these methods primarily address static editing, suitable for one-time modifications, and struggle to accommodate sequential editing demands over time. To address these limitations,

GRACEHartvigsen et al. (2023) replace hidden states with vectors retrieved from a learned code-book. MELOYu et al. (2024) introduces a vector database and leverages a clustering mechanism to dynamically assign LoRA modules. ELDERLi et al. (2025) adopts a MoE framework, where a learnable neural network weights shared LoRA modules. However, these approaches inevitably update the semantic representation in sequential edits, leading to semantic drift or knowledge forgetting.

# 3 METHOD

## 3.1 PRELIMINARIES

**Lifelong Model Editing** The purpose of lifelong model editing, as described by GraceHartvigsen et al. (2023), is to make continual edits to the knowledge of the initial base model without degrading the performance of the base model and without negating previous edits. Consider an initial base model $f_{base}$, After $n$ edits $D_{edit} = \{d_1, ..., d_n\}$, some of the model's knowledge is updated and model is transformed into $f_n$, where $d_i = (\boldsymbol{x}_i, \boldsymbol{y}_i)$, For lifetime model editing, the edited model $f_n$ should be able to correctly output $\boldsymbol{y}$ within the edited inputs $\boldsymbol{x} \in D_{edit}$, i.e., $f_n(\boldsymbol{x}_i) = \boldsymbol{y}_i$. Furthermore, for inputs $\boldsymbol{x}' \notin D_{edit}$ that are semantically similar to the edited instances, such as rephrased sentences, the model $f_n$ is expected to generalize the updated knowledge appropriately, like $f_n(\boldsymbol{x}'_i) = \boldsymbol{y}_i$. Moreover, after editing, the model $f_n$ should also retain the previous knowledge of $f_{base}$, that is, for data samples $\boldsymbol{x}_j \notin D_{edit}$ that have not been edited, there should be $f_n(\boldsymbol{x}_j) = f_{base}(\boldsymbol{x}_j) = \boldsymbol{y}_j$.

**LoRA for Lifelong Model Editing** Low-Rank Adaptation (LoRA) Hu et al. (2022) is an efficient finetuning method for LLMs that optimizes model outputs by projecting features into a low-rank subspace. In the context of efficient LLM finetuning, LoRA modules are typically inserted into specific layers of the pre-trained model. During editing, the base LLM parameters $W_0 \in \mathbb{R}^{d \times k}$ remain frozen, while only the LoRA module $\Delta W \in \mathbb{R}^{d \times k}$ is updated. The parameter update $\Delta W$ can be represented as a low-rank decomposition of the pretrained weight matrix. Specifically, $\Delta W$ is factorized into two smaller matrices $A \in \mathbb{R}^{r \times k}$ and $B \in \mathbb{R}^{d \times r}$, such that $\Delta W = BA$, where $r \ll \min(d, k)$ and $r$ is the LoRA rank. Given the original $\boldsymbol{h} = W_0\boldsymbol{x}$, the forward propagation process is modified as:

$$\boldsymbol{h} = W_0\boldsymbol{x} + \Delta W\boldsymbol{x} = W_0\boldsymbol{x} + BA\boldsymbol{x} \tag{1}$$

where $A$ is initialized using a zero mean Gaussian distribution and $B$ is initialized as a zero matrix. This approach significantly reduces the number of trainable parameters while maintaining the expressive power of the full-rank update.

## 3.2 CLUSTER UPDATING IN LIFELONG MODEL EDITING

In the context of lifelong model editing, when multiple adapter modules are involved, the model needs to dynamically manage the association between edits and the corresponding adapter module. MELO addresses this by employing a clustering mechanism that assigns edits to clusters based on the hidden representations of the inputs, with cluster centres being continuously updated throughout editing. However, this continual updating of cluster centers alters their semantic representation, leading to semantic drift and incorrect module assignments. Moreover, repeated training of LoRA modules can result in forgetting previously acquired knowledge. Varying the cluster radius in MeLO results in a different number of generated cluster centres, thereby affecting the frequency of cluster centre updates. We record MeLO's performance under different numbers of cluster centre updates, as shown in Fig.1b. The experiment is conducted on SCOTUS dataset. In Fig.1b, the number of updates refers to the number of cluster centre updates during the editing process, and the number of mismatches indicates how many times, during inference, the retrieved LoRA module differs from the one assigned during editing. ERR and TRR represent the model's accuracy on the edited and unedited datasets, respectively, after all editing tasks are completed.

As shown in the Fig.1b, for the same datasets SCOTUS of editing, increasing the number of cluster centre updates leads to a rise in mismatches during inference, and a corresponding decline in model performance on both the edited and unedited datasets. This demonstrates that cluster centre up-

dates can cause semantic drift, resulting in incorrect LoRA match, while repeated training of LoRA modules contributes to knowledge forgetting. Therefore, in SoLA, we freeze both the LoRA modules and their associated keys once the current task is completed. These components are no longer updated in subsequent edits, thereby maximizing knowledge retention across editing iterations.

Table 1: Comparison of SoLA to existing methods. All the results are obtained after sequential edits. ERR indicates Edit Reliability Rate, TRR presents Task Retention Rate, and ARR is Accuracy Attention Rate. The best performance is shown in bold.

| Method | SCOTUS (BERT; Acc ↑) | | | zsRE (T5; F1 ↑) | | | Hallucination (GPT2-XL; PPL ↓) | | |
|---|---|---|---|---|---|---|---|---|---|
| | ERR | TRR | $Avg.$ | ERR | TRR | $Avg.$ | ERR | TRR | ARR |
| EWC | 0.51 | 0.82 | 0.66 | 0.67 | 0.50 | 0.58 | 1485.7 | 29.24 | 109.59 |
| CMR | 0.52 | 0.52 | 0.52 | 0.56 | 0.82 | 0.69 | 1449.3 | 28.14 | 107.76 |
| CLEAR | 0.67 | 0.83 | 0.75 | 0.27 | **0.99** | 0.63 | 2394.3 | 35.34 | 195.82 |
| MEND | 0.19 | 0.27 | 0.23 | 0.25 | 0.27 | 0.26 | 1369.8 | 1754.9 | 2902.5 |
| SERAC | 0.33 | 0.41 | 0.37 | 0.72 | 0.31 | 0.52 | 8183.7 | 133.3 | 10.04 |
| ROME | - | - | - | - | - | - | 30.28 | 103.82 | 14.02 |
| GRACE | 0.81 | 0.82 | 0.82 | 0.69 | 0.96 | 0.83 | 15.84 | 7.14 | 10.00 |
| ELDER | 0.89 | 0.86 | 0.88 | 0.72 | 0.97 | 0.84 | 16.12 | 5.87 | 8.42 |
| MELO | 0.96 | 0.92 | 0.94 | 0.72 | 0.98 | 0.85 | 17.45 | 1.04 | **2.66** |
| SoLA | **0.97** | **0.95** | **0.96** | **0.73** | **0.99** | **0.86** | **15.15** | **1.01** | 7.35 |

### 3.3 SEMANTIC ROUTING-BASED CONTROLLABLE LoRA

In the layer where the model needs to be edited, we keep the parameters of the original model frozen and insert the learnable LoRA module for editing, as shown in Fig.2 a). In SoLA, each editing operation is encapsulated within an independent LoRA module. During sequential editing, for a given editing task $d_i = (\boldsymbol{x}_i^m, \boldsymbol{y}_i^m)_{m=1}^n$, a dedicated LoRA module LoRA$_i$ is assigned. For each data instance $\boldsymbol{x}_i^m \in d_i$, the corresponding LoRA id $i$ is recorded, and a semantic routing entry is established by associating LoRA module with a semantic key derived from the input representation, as shown in Fig.2 b). Following prior workHartvigsen et al. (2023), we use the hidden representation of the last token in the input sequence as the key $\boldsymbol{e} \in \mathbb{R}^d$ vector. During editing, the assigned LoRA module LoRA$_i$ will fully learn the task $d_i$, yielding the output:

$$\boldsymbol{h} = \boldsymbol{h_0} + \text{LoRA}_i(\boldsymbol{x}) \tag{2}$$

where $\boldsymbol{h_0} \in \mathbb{R}^{l \times d}$ is the frozen base model representation, $l$ is the sequence length, $d$ is the hidden state dimension. At this stage, all other LoRA modules and the stored key vectors remain frozen. Upon completion of the editing process, the trained LoRA$_i$ module is frozen and will be stored with its associated key $\boldsymbol{e}_i$ as a fixed mapping for future reference. Neither the LoRA module nor the key will be updated in subsequent editing. Since only the current LoRA module is involved in training during each edit, SoLA significantly reduces the number of trainable parameters, as shown in Fig.1a. Furthermore, freezing both the LoRA module and its corresponding key after editing prevents semantic drift and mitigates catastrophic forgetting, thereby maximizing knowledge retention. During inference, the hidden representation of the last token in the input sequence will serve as a query vector $\boldsymbol{q} \in \mathbb{R}^d$ from the input $\boldsymbol{x}$ and matches against the stored keys $K \in \mathbb{R}^{N \times d}$, as shown in Fig.2 c). If a match is found, the corresponding LoRA module is retrieved from stored LoRA pool $R \in \mathbb{R}^M$ and incorporated into the computation.

### 3.4 MASTER DECISION MECHANISM

During inference, existing approaches usually need to introduce an auxiliary routing network outside the target edited layer to decide whether to activate the LoRA module or notYu et al. (2024)Li et al. (2025), which hampers the end-to-end decision-making process. To address this limitation, we propose a master decision-making mechanism that does not require an auxiliary routing component. Specifically, we designate the first edited layer as the master decision layer where input features are dynamically evaluated to activate the relevant LoRA module without the need for an auxiliary

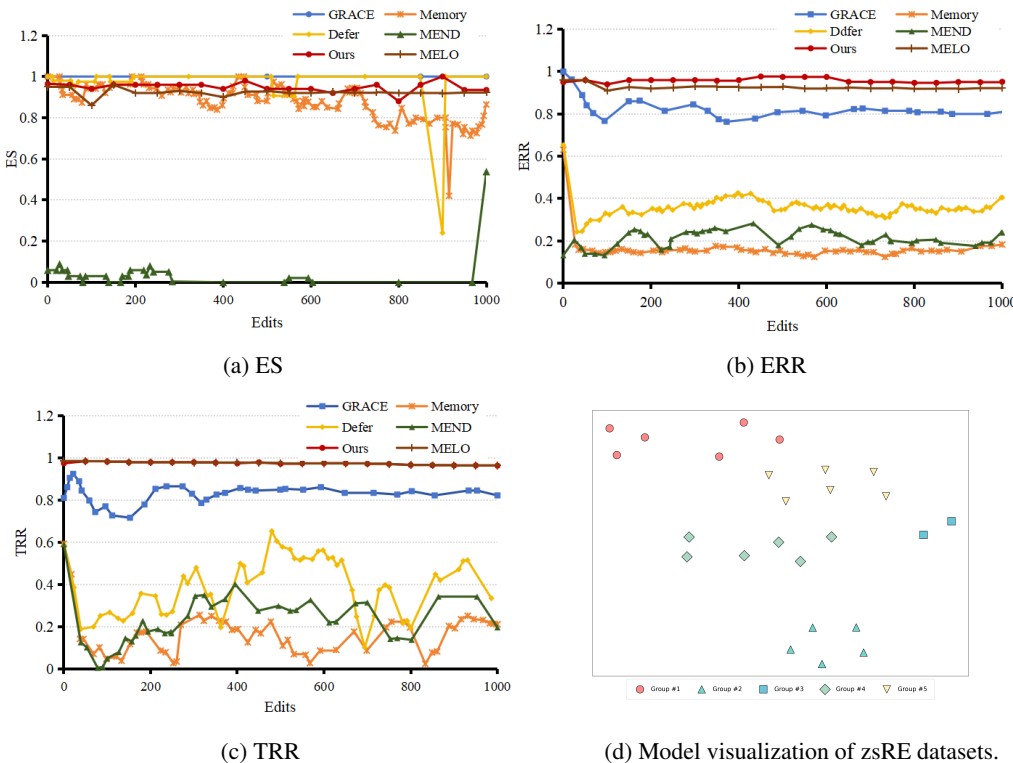

Figure 3: Figure(a–c) presents every edit accuracy on SCOTUS datasets with BERT. Figure(d) shows visualization of zsRE dataset encoder output from T5-small with t-SNE and the dots with same color and shape are input and rephrase sentence. All experiment settings are the same as Tab. 1.

network. In this framework, the master decision layer $H_e$ computes the distance metric between the input feature embedding $q$ and the stored key $K$ to determine the activation of the module: $d = H_e(argmin_i(\text{dist}(q, k_i))), i = (1, 2, ..., N)$, dist($\cdot$) denotes the distance function and each key $k_i$ associated with the LoRA module. Then in the master layer $H_e$, we decide the edit layer behaviour:

$$H_e = \begin{cases} H(W_0) & \text{if } d \geq \alpha \\ H(W_0, W_{R_m}) & \text{if } d < \alpha \end{cases} \tag{3}$$

where $\alpha$ is the threshold, in this work we set it as 0.01, $W_{R_m}$ represents the weight of $m$-th LoRA module associated with the key nearest to the query. This binary decision is propagated to the subsequent edited layers, ensuring consistent module activation throughout the network. By integrating the decision-making process to the first edited layer, our approach achieves complete end-to-end decision-making capability while maintaining architectural simplicity.

## 4 EXPERIMENTS

### 4.1 IMPLEMENTATION DETAILS

**Baselines** We first compare several recent methods designed for lifelong model editing. GraceHartvigsen et al. (2023) replaces model outputs with key-value pairs and matches features to keys based on deferral radius. MELOYu et al. (2024) leverages LoRA for finetuning and dynamically retrieves relevant LoRA modules via neuron index. ELDERLi et al. (2025) employs MoE to dynamically combine multiple LoRA modules. Additionally, we evaluate other baseline methods. MENDMitchell et al. (2021) uses a hypernetwork trained on auxiliary data to predict the required gradients for model editing. SERACMitchell et al. (2022) learns a scope classifier and a counterfactual model, coordinating between modules to determine editing behavior. ROMEMeng et al. (2022a)

identifies the most influential weights in the model through perturbation analysis, localizes knowledge to specific layers of GPT, and modifies the corresponding weights. Since ROME is specifically designed for GPT, it is only evaluated on the hallucination correction task. EWC(Kirkpatrick et al., 2017) imposes constraints on the weights updating so that the model continuously learns new knowledge and retains previous knowledge. CMRLin et al. (2022) performs continuous finetuning of the model to continuously learn new knowledge. CLEARRolnick et al. (2019) builds a memory buffer to restore old tasks information, and replays them in subsequent edits to retain previous knowledge.

**Evaluation Metric** In this work, followingHartvigsen et al. (2023), we use three basic metrics to assess model performance: ES, ERR and TRR, which are applicable to all model experiments. Where Edit Success(ES) measures the ability of the model to edit the target sample. Edit Reliability Rate(ERR) assesses the validity of the model on the edited dataset, indicating the extent to which the required knowledge updates are successfully incorporated. Task Retention Rate(TRR) quantifies the model's performance on unedited datasets and reflects the model's ability to retain previous knowledge. Additionally, in the hallucination correction task, we also use the Accuracy Attention Rate (ARR) to assess the performance of the already accurate output. Accuracy is measured differently for different tasks. For the document classification task of SCOTUS, we use the Average Accuracy Rate (ACC); for the question answering task of zsRE, we measure it using the average F1, and for the hallucination correction task, we measure it using the standard average complexity (PPL)Brown et al. (1992).

Table 2: Model Performance Comparison on UniEdit and WikiBigEdit datasets in hallucination correction tasks. All the scores are PPL metric and the best result is in bolded.

| Model | Method | UniEdit (PPL ↓) | | | WikiBigEdit (PPL ↓) | | |
|---|---|---|---|---|---|---|---|
| | | ERR | TRR | ARR | ERR | TRR | ARR |
| LLaMA-3-8B | EWC | 2.43 | 1134 | **22.08** | 3.58 | 251117 | 4.93 |
| | CMR | 2.79 | 2239 | 25.24 | 2.86 | 169452 | 4.49 |
| | CLEAR | 1.04 | 982 | 29.79 | 1.03 | 74384 | **1.52** |
| | SERAC | 37.10 | 246 | 93 | 60.17 | 215 | 59 |
| | GRACE | **1.00** | 244 | 89 | **1.00** | 216 | 58 |
| | ELDER | **1.00** | 173 | 74 | **1.00** | 178 | 48 |
| | MELO | **1.00** | 153 | 68 | **1.00** | 165 | 47 |
| | Ours | **1.00** | **144** | 66 | **1.00** | **162** | 44 |
| DeepSeek-R1-8B | EWC | 2.38 | 4451 | 42.13 | 2.86 | 47846 | 3.89 |
| | CMR | 2.34 | 7211 | **36.19** | 3.60 | 34623 | 5.51 |
| | CLEAR | 1.06 | 3365 | 44.86 | 1.02 | 28110 | **1.48** |
| | SERAC | 46.76 | 451 | 834 | 82.72 | 493 | 831 |
| | GRACE | **1.00** | 407 | 803 | **1.00** | 487 | 793 |
| | ELDER | 1.03 | 241 | 482 | 1.03 | 304 | 492 |
| | MELO | 1.09 | 204 | 407 | 1.05 | 263 | 445 |
| | Ours | 1.07 | **188** | 374 | **1.00** | **248** | 412 |
| Qwen2-7B | EWC | 4.62 | 7648 | 48.62 | 5.61 | 61834 | 14.65 |
| | CMR | 4.12 | 11583 | **40.56** | 6.28 | 48629 | 16.89 |
| | CLEAR | 1.03 | 5021 | 54.98 | 1.05 | 35396 | **4.26** |
| | SERAC | 58.51 | 329 | 1261 | 62.12 | 304 | 523 |
| | GRACE | 2.05 | 316 | 1065 | 16.88 | 299 | 518 |
| | ELDER | **1.00** | 201 | 266 | **1.00** | 225 | 197 |
| | MELO | **1.00** | 170 | 201 | **1.00** | 203 | 143 |
| | Ours | **1.00** | **156** | 110 | **1.00** | **199** | 109 |

## 4.2 MAIN RESULTS

We evaluate the performance of the proposed SoLA method on three benchmark datasets and compare it with several state-of-the-art model editing approaches. The results are summarized in Tab.1. As observed, SoLA achieves excellent performance on most of the benchmark datasets, with SoLA outperforming the strongest baseline, MELO, by 3% on the SCOTUS dataset. These demonstrate SoLA's enhanced effectiveness in editing target knowledge while better preserving previously acquired knowledge. Furthermore, existing methods such as MEND and SERAC perform poorly under the sequential editing setting, suggesting that they are primarily designed for static editing scenarios and are not well-suited for sequential model editing tasks.

We also report task-wise accuracy progression to analyze model performance throughout the sequential editing process. The experimental results are based on the SCOTUS dataset and the results are shown in Fig.3. As illustrated in Fig.3a, Fig.3b and Fig.3c, SoLA consistently maintains superior accuracy across tasks, without exhibiting substantial performance degradation or volatility. This indicates the robustness and stability of SoLA when applied to sequential knowledge editing.

Method performance may vary across different model sizes and datasets. To further evaluate the robustness of our approach, we conduct hallucination correction experiments on the UniEditChen et al. (2025) and WikiBigEditThede et al. (2025) datasets using larger-scale backbone models. The results are presented in Tab.2. As shown in Tab.2, SoLA consistently achieves the best or near-best performance across most settings, demonstrating its robustness and effectiveness under diverse task configurations. During editing, larger backbone models generally exhibit more stable ERR values, suggesting that such models have learned stronger semantic representations during pretraining, leading to better generalization and editability. Furthermore, continuous learning related methods tend to overfit in edited data, thus showing better ARR in the retention set, but large TRR values in the upstream set, indicating severe forgetting of previous knowledge.

## 4.3 CONTROLLABLE ROLLBACK EDITING

In SoLA, each edit instance is associated with a unique key stored in both the memory and the routing table. This design enables reversible editing, as the model can revert to its original behavior by removing the key corresponding to a specific edit. To demonstrate this property, we conduct an illustrative experiment on the zsRE dataset. The results are shown in Tab.3. In the Tab.3, each row corresponds to an input instance consisting of a "Text" (the question need to be edited) and its "Labels" (the correct answer). "Del" indicates whether the key corresponding to the edit is removed after editing. For each input, $Pred_{base}$, $Pred_{edit}$ and $Pred_{del}$ denote the prediction of base model, edited model and model after removing the corresponding key. When "Del" is false, the key is retained, serving as a baseline comparison. As shown in Tab.3, for each instance, the original prediction ($Pred_{base}$) does not match the correct label, indicating the necessity for editing. After editing, the model correctly produces $Pred_{edit}$ consistent with the ground-truth label, demonstrating that SoLA successfully updates the model's knowledge. When the associated key is deleted, the model reverts to its original prediction ($Pred_{edit} = Pred_{base}$), verifying that our method can effectively roll back specific edits. More critically, for edits where the key is not deleted, the model continues to output $Pred_{edit}$, showing that the rollback operation does not interfere with other edits. This confirms that SoLA supports fine-grained, selective undo of knowledge edits without disrupting unrelated modifications.

Table 3: Controllable edit in SoLA on zsRE datasets. "Text" is the question need to be edited, "Labels" is the true answer, "Del" indicates whther delete associated key and $Pred_{base}$, $Pred_{edit}$ and $Pred_{del}$ is the prediction of base model, edited model and delted model.

| Text | Labels | Del | $Pred_{base}$ | $Pred_{edit}$ | $Pred_{del}$ |
|------|--------|-----|---------------|---------------|--------------|
| The date of Tsetserleg earthquake in 1905? | 9 July 1905 | True | 20 February 1905 | 9 July 1905 | 20 February 1905 |
| What constellation has Nu Cancri? | Cancer | True | Hydra | Cancer | Hydra |
| On what date did the Bahrain Grand Prix 2015 occur? | 19 April 2015 | True | 6 April 2017 | 19 April 2015 | 6 April 2017 |
| What position does Charles-Joseph Coursol have? | Mayor of Montreal | True | Positioning | Mayor of Montreal | Positioning |
| Who acted in Blow Dry? | Alan Rickman | False | Jim Carrey | Alan Rickman | Alan Rickman |

## 4.4 ABLATION STUDY

**Effect of Edit Layer Location** In semantic representation learning, different layers of a model focus on different aspects of semantic information. Therefore, the position of the edited layer can significantly influence the effectiveness of model editing. To investigate this, we evaluate the performance of model when edits are applied at different layers, while keeping all other settings fixed. The experiments are conducted on the SCOTUS dataset with a Nvidia A100 40G GPU, and the results are shown in Tab.4a. In the Tab.4a, "Layer" denotes the layer range used for editing. For example, "0–2" indicates that edits are applied to layers 0, 1, and 2 of the BERT model. As shown in the table, editing at shallower layers results in suboptimal performance. This observation aligns with prior findingsGeva et al. (2020), which suggest that shallow layers primarily capture the shallow sentence patterns, whereas deeper layers encode richer semantic features, thereby enabling more ef-

fective knowledge editing. Moreover, editing at shallow layers significantly increases the time used for editing, which means that editing for semantics is more difficult in shallow layers. Furthermore, we observe that varying the editing layer has negligible impact on performance over the unedited portion of the dataset, indicating that SoLA is capable of preserving previously learned knowledge regardless of the editing depth, which further supports the robustness of SoLA.

Table 4: Ablation studies on (a) location of edited layers and (b) LoRA rank. Here, "Layer" indicates the edited layer location. For example, "0-2" presents edited layer is layers 0, 1 and 2. "LoRA Rank" indicates the rank value of every LoRA in model.

(a) Effect of location of edited layer.

| Layer | ES | ERR | TRR | Edit Time(min) |
|-------|------|------|------|----------------|
| 0-2 | 0.77 | 0.61 | 0.96 | 9.21 |
| 3-5 | 1.00 | 0.68 | 0.96 | 8.06 |
| 6-8 | 0.81 | 0.70 | 0.97 | 7.03 |
| 9-11 | 0.94 | 0.95 | 0.97 | 5.97 |

(b) Effect of LoRA rank.

| LoRA Rank | ES | ERR | TRR | Edit Time(min) |
|-----------|------|------|------|----------------|
| 1 | 0.90 | 0.84 | 0.96 | 5.88 |
| 2 | 0.94 | 0.91 | 0.97 | 5.90 |
| 3 | 1.00 | 0.91 | 0.97 | 5.86 |
| 4 | 1.00 | 0.95 | 0.97 | 5.97 |
| 5 | 1.00 | 0.92 | 0.96 | 5.86 |
| 10 | 1.00 | 0.71 | 0.96 | 5.85 |

**Effect of LoRA Rank** The LoRA module projects semantic representations into a low-rank subspace for parameter-efficient learning. Consequently, the choice of rank will effect the module's performance. To investigate this, we analyze the impact of different LoRA rank values on model performance, keeping all other settings fixed. Experiments are conducted on the SCOTUS dataset using a NVIDIA A100 40GB GPU, and the results are presented in Tab.4b. As shown in the Tab.4b, simply increasing the LoRA rank does not lead to improved performance and may even degrade model performance. This indicates that expanding the learnable parameter space does not necessarily enhance the model's capacity for effective knowledge editing. On the contrary, excessive rank values may even result in performance degradation due to overfitting, suggesting the importance of carefully selecting an appropriate rank to balance capacity and generalization. Based on the experiment results, we set 4 as LoRA rank for all the experiments in this paper.

### 4.5 MODEL VISUALIZATION

To gain deeper insight into the model's behavior, we conduct a t-SNE visualization Maaten & Hinton (2008) of the learned feature representations. The experiment is conducted on the zsRE dataset using the T5-small model . Specifically, after all whole editing tasks are completed on question answering task, we select five input instances along with their rephrase sentence as 5 group, and extract the encoder output features from the edited T5-small model for visualization. The resulting plot is shown in Fig.3d. As illustrated in the Fig.3d, SoLA is able to encode semantically similar inputs into nearby representations, indicating that it effectively captures and preserves semantic information while maintaining meaningful relative distances in the representation space. This supports the model's ability to generalize edits based on semantic similarity.

## 5 CONCLUSION

In this paper, we propose SoLA, a Semantic routing-based LoRA framework for reversible lifelong model editing. In SoLA, each edit is encapsulated as an independent LoRA module which is frozen after training and a semantic routing record is established to map LoRA module to the input semantic representation, allowing dynamic activation of LoRA modules via semantic matching. This design not only avoids semantic drift caused by clustering updating but also mitigates catastrophic forgetting from parameter sharing. More importantly, SoLA supports precisely revoking edit by removing the corresponding key from the routing table, enabling reversible model editing. This allows for flexible addition and deletion of edits, offering fine-grained control over model behavior. To the best of our knowledge, this is the first work to achieve reversible model editing in the literature. Furthermore, we introduce a master decision-making mechanism by integrating decision-making into the edited layer, enabling end-to-end decision-making process. By building a strict mapping between edits and LoRA modules, SoLA achieves efficient and reversible lifelong model editing, providing a novel perspective for future research.

ETHICS STATEMENT

Our work presents a novel algorithm in the field of Lifelong Model Editing. We are not aware of any direct ethical issues arising from this research, as it does not involve human subjects, private data, or foreseeable immediately harmful applications. We have reviewed and adhered to the ICLR Code of Ethics.

REPRODUCIBILITY STATEMENT

- **Code:** Upon acceptance of this paper, the source code will be made publicly available.
- **Experiments:** The complete experimental setup, including hyperparameter values and training details, is described in Section 4.
- **Data:** All experiments in this paper were performed on the (**publicly available benchmark dataset**. A complete list of the datasets used and their respective citations can be found in Section 4.

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

# Appendix

## A  THE USE OF LLM

In accordance with ICLR 2026's policy on the use of Large Language Models (LLMs), we disclose that we used OpenAI GPT-4o for writing embellishment. The model was prompted to "help me polish this paragraph, using academic style of expression." the LLM was used for language touch-ups only and did not generate any new content, experimental results, or analysis. All results were reviewed and verified by the authors, who are solely responsible for the final manuscript.

## B  ADDITIONAL RELATED WORK

### B.1  PARAMETER-EFFICIENT FINE-TUNING

Parameter-efficient fine-tuning (PEFT) approaches adjust pretrained LLMs by incorporating lightweight modules, while keeping the base model frozen. Optimization is performed solely through updates to these lightweight components. Representative approaches include AdaptersHoulsby et al. (2019)Zaken et al. (2021), PromptsLi & Liang (2021)Jia et al. (2022), and Low-Rank Adaptation (LoRA)Hu et al. (2022)Valipour et al. (2022). Adapters are compact bottleneck modules located after transformer blocks; Prompts are learnable vectors prepended to the input sequence; and LoRA uses low-rank decomposition matrices to modify model weights during training. Recent progress explores applying PEFT to lifelong model editing, where the base model remains unchanged and new parameter modules are introduced to manage sequential edits over time.

### B.2  ROUTING MECHANISMS IN MODULAR LANGUAGE MODELS

When multiple modular components (e.g., LoRA modules) are present, a key challenge lies in routing input queries to the appropriate subset of modules during inference. Recent work explore various strategies to support dynamic module selection. MELOYu et al. (2024) introduces a clustering mechanism to assign edits to clusters based on input hidden states. During inference, the nearest cluster center is used to retrieve relevant LoRA modules. However, cluster centers are inevitably updated during the sequential editing, which may lead to semantic drift and incorrect matching. As shown in Fig.1b, the number of incorrect matches increases significantly as the number of cluster centre updating increases, which leads to a deterioration in model performance. ELDERLi et al. (2025) employs MoE approach in which a learnable neural network scores a set of LoRA modules, activating the top-k modules. This strategy offers parameter efficiency but introduces soft entanglement in routing, and due to shared module usage, new edits may interfere with or overwrite previously edits. Furthermore, both strategies rely on auxiliary routing networks, adding architectural complexity and computational overhead. They also offer limited control over edit selection, making rollback and deletion of edits challenging.

## C  ADDITIONAL EXPERIMENT DETAILS

**Benchmarks** We evaluate the performance of the model in three representative tasks: Document Classification, Question Answering(QA), and Hallucination Correction. Among them, the document classification task is performed based on the SCOTUS datasetChalkidis et al. (2022). SCOTUS is a subset of Fairlex, which collects documents from the U.S. Supreme Court and classifies them into 11 topics. In practice, the topics corresponding to the documents will change, so it is necessary to update the model knowledge to classify the documents into new topics. We divide SCOTUS into an edit set and an upstream set, where models are edited on the edit set and the upstream set is not involved in editing, only testing. The question answering task is then conducted on the zsRE datasetLevy et al. (2017), and NQ datasetsKwiatkowski et al. (2019) serves as upstream dataset. Specifically, we perform editing in zsRE, and the NQ dataset is again not involved in editing, only testing. For hallucination correction, we adopted the framework introduced byManakul et al. (2023),

aiming to address the tendency of GPT models to generate factually inconsistent outputs. Following GRACEHartvigsen et al. (2023), we employ SelfCheckGPTManakul et al. (2023) for editing. SelfCheckGPT is a Wikipedia-style biography that is first generated using GPT3 on 256 topics, e.g., "Bon Jovi", and then checked to see which of the generated biographies are hallucinations. SelfCheckGPT contains 1392 hallucinatory sentences to be edited, which serves as the editing set, and 516 already correct sentences as the retention set, which is used to measure model's post-edit perplexity on the already correct sentences. WebTextNakano et al. (2021) is used as the upstream set, remaining unedited and reserved for testing. In addition to this, in order to test the performance of the model approach with different sized models as well as datasets, we conducted hallucination correction experiments on UnieditChen et al. (2025) and WikiBigEditThede et al. (2025) to evaluate the generalization performance of the models. Uniedit is based on open domain knowledge covering information from 25 common domains in 5 major categories. The WikiBigEdit dataset is based on regular updates of the knowledge graph in Wikipedia and contains 5 months of extensive factual editing and improvement. The Uniedit and WikiBigEdit datasets already have edit set, retention set and upstream set that can be used directly for hallucination correction tasks.

**Training details** FollowingHartvigsen et al. (2023), we employ BERT, T5-small, and GPT2-XL as the base models for the three tasks—document classification, question answering, and hallucination correction, respectively. Among these, BERT and T5-small are the official pre-trained models, while GPT2-XL is obtained from Hartvigsen et al. (2023) which finetuned from the official releaseHartvigsen et al. (2023). For training, we adopt Stochastic Gradient Descent (SGD) as the optimizer with a cosine decay learning rate schedule. The learning rates is 0.05, training epochs is 40, and LoRA rank is 4. In the three tasks, we edit the models and the edited layers of the individual model edits are shown in the Tab.5.

Table 5: Edited layers of different model

| Model | Edit layer |
|---|---|
| BERT | bert.encoder.layer.9.output.dense |
| | bert.encoder.layer.10.output.dense |
| | bert.encoder.layer.11.output.dense |
| T5-Small | encoder.block.5.layer.1.DenseReluDense.wi_0 |
| | encoder.block.5.layer.1.DenseReluDense.wi_1 |
| | encoder.block.5.layer.1.DenseReluDense.wo |
| | encoder.block.6.layer.1.DenseReluDense.wi_0 |
| | encoder.block.6.layer.1.DenseReluDense.wi_1 |
| | encoder.block.6.layer.1.DenseReluDense.wo |
| GPT2-XL | transformer.h.36.mlp.c_fc |
| | transformer.h.37.mlp.c_fc |

Table 6: Experiments results on long sequential edits(a) and different $\alpha$ values(b). The best results are shown in bold.

(a) Performance comparison of models on the long sequential edits(5000 edits).

| | zsRE (T5; F1 ↑) | | |
|---|---|---|---|
| Methods | ERR | TRR | $Avg.$ |
| GRACE | 0.66 | 0.94 | 0.80 |
| MELO | 0.71 | 0.97 | 0.84 |
| SolA | **0.73** | **0.98** | **0.86** |

(b) Model performance at different $\alpha$ values on the zsRE datasets.

| | zsRE (T5; F1 ↑) | | |
|---|---|---|---|
| $\alpha$ | ERR | TRR | $Avg.$ |
| 0.01 | 0.73 | 0.99 | 0.86 |
| 0.1 | 0.73 | 0.99 | 0.86 |
| 1 | 0.73 | 0.99 | 0.86 |
| 5 | 0.72 | 0.99 | 0.85 |
| 10 | 0.71 | 0.99 | 0.85 |

# D  ADDITIONAL EXPERIMENT RESULTS

To evaluate the model's performance in long sequence editing scenarios, we conducted tests involving 5000 edits, with the results summarized in Tab.6a. As shown in Tab.6a, SoLA continues

to outperform other methods in long sequence editing tasks, demonstrating its robust capability to maintain superior performance even under extended editing sequences.

Furthermore, to assess the sensitivity of the model to the threshold parameter $\alpha$, we examined its performance across different $\alpha$ values, as presented in Tab.6b. The results indicate that SoLA maintains consistent stability across varying $\alpha$ values, further confirming its strong robustness.

