# OpenReview forum: "Reversible Lifelong Model Editing via Semantic Routing-Based LoRA"
_ICLR.cc/2026/Conference — ICLR 2026 Conference Withdrawn Submission_

### Official Review · Reviewer_3fBn · 2025-10-24

**Soundness:** 2
**Presentation:** 3
**Contribution:** 3
**Rating:** 6
**Confidence:** 4

**Summary:**

This paper addresses catastrophic forgetting and semantic drift in lifelong model editing. The authors propose SoLA, a framework that assigns a separate, parameter-efficient LoRA module to each knowledge edit. After training, the LoRA module and its corresponding semantic "key" (generated from the input representation) are frozen. During inference, a semantic routing mechanism matches the input "query" with stored "keys" to dynamically activate the relevant LoRA module. The key innovation is enabling reversible edits: an edit can be precisely undone by removing its "key" from the routing table, without retraining. Experiments on document classification, question answering, and hallucination correction demonstrate SoLA's effectiveness in achieving accurate, efficient, and reversible editing.

**Strengths:**

Strength 1: The proposed "reversible editing" mechanism is a significant and novel contribution to the field of model editing. In real-world applications requiring frequent model knowledge updates and corrections, the ability to precisely and efficiently undo erroneous or outdated edits is highly valuable.

Strength 2: Effective mitigation of catastrophic forgetting

**Weaknesses:**

Weakness 1: I see a scalability issue, and this is the method's primary theoretical and practical limitation. The design implies that auxiliary parameters grow linearly and indefinitely with the number of edits. Although each LoRA module is small, in a true **lifelong learning** scenario involving millions of updates, this approach incurs significant storage and management overhead. This raises concerns about long-term scalability, especially compared to Mixture-of-Experts methods that use a fixed parameter pool, representing a clear drawback. I encourage the authors to conduct a quantitative analysis of this issue to properly assess the method's performance in more challenging scenarios (e.g., 3k–100k edits).

Weakness 2: Another issue is the limitation in knowledge generalization, such as hierarchical reasoning and attribute inheritance. The proposed design with independent LoRA modules may struggle in these aspects. I encourage the authors to present relevant case studies and analysis.

**Questions:**

See weaknesses

---

> ### Author Response · Authors · 2025-11-23
>
> Dear reviewer, we thank you for your positive assessment of our contributions, particularly your recognition that reversible editing is ***"a significant and novel contribution"*** and your affirmation of the effective mitigation of the forgetting issue.
>
> >**1. Scalability issue**
>
> We thank the reviewer for raising the point regarding SoLA's linear storage growth with the number of edits. The core contribution of SoLA lies in achieving precise and reliable reversible lifelong editing. In this work, we consciously prioritized editing stability and reversibility, accepting a manageable increase in storage (relative to the base model) as a deliberate design trade-off to realize these capabilities.
>
> Besides, following your suggestion, we have extended the experiments to `5000 edits`, as shown in the table below. The results demonstrate that SoLA still significantly outperforms other methods in long-sequence editing, confirming its robustness and sustainability over extended editing processes. We fully agree that scalability is a crucial direction for lifelong learning systems and consider this a key area for future optimization, such as exploring lossless or low-loss merging and compression strategies for historical LoRA modules.
>
> || zsRE (T5; F1 ↑) | | |
> | :--------------- | :-------------: | :-: | :-:|
> | Methods| ERR | TRR| Avg |
> | GRACE| 0.66| 0.94 | 0.80 |
> | MELO| 0.71| 0.97| 0.84 |
> | SoLA| **0.73** | **0.98** | **0.86** |
>
> >**2. The limitation in knowledge generalization**
>
> You have raised a highly valuable point regarding the limitations in knowledge generalization, such as hierarchical reasoning and attribute inheritance. This is an important insight for advancing research in lifelong model editing. Addressing this will require redesigning the experiments, and we will conduct these studies in the future. We sincerely appreciate your insightful comments, which have pointed out a key direction for our future research.

---

> ### Author Response · Authors · 2025-11-28
>
> Dear reviewer, we hope you are doing well. We would like to kindly follow up regarding the clarifications and additional large-scale experiments we provided in our rebuttal, especially on the key points you raised about scalability under long editing sequences and the limitations in knowledge generalization. We fully understand you may be handling many reviews at this stage, but we wanted to ask whether there is any further information or analysis we can provide that would help address your concerns.
>
> We sincerely appreciate your recognition of reversible editing as ***“a significant and novel contribution”*** and your constructive feedback on long-term scalability and generalization. If there are any remaining issues you would like us to elaborate on, we would be more than happy to assist.
>
> Thank you again for your time and consideration, and we look forward to any additional comments you may have.

---

### Official Review · Reviewer_ucAF · 2025-10-29

**Soundness:** 3
**Presentation:** 3
**Contribution:** 2
**Rating:** 4
**Confidence:** 3

**Summary:**

This paper proposes a continual model editing method that enables controllable knowledge editing by storing a semantic key and a corresponding LoRA for each sample that requires editing.

**Strengths:**

- The method proposed in this paper allows more controllable addition and removal of knowledge during model editing.
- The paper is well-organized and structured.

**Weaknesses:**

- The proposed method requires adding a key and a separate LoRA for each sample to be edited, and these parameters are frozen and stored together with the model. Therefore, as the number of samples to be edited increases, the total number of model parameters will grow significantly.
- This paper only compares the number of parameters involved during editing, but lacks a comparison of the total model parameters for each method.
- It does not discuss and compare important methods in the field, such as AlphaEdit [1].
- The paper edits only 1,000 samples (as shown in Figure 3). However, methods in this field typically use a larger number of edits — for example, AlphaEdit performs edits on 3,000 samples.

[1] Fang, Junfeng, et al. "AlphaEdit: Null-Space Constrained Knowledge Editing for Language Models." The Thirteenth International Conference on Learning Representations 2025.

**Questions:**

Refer to the Weaknesses section

---

> ### Author Response · Authors · 2025-11-23
>
> Dear reviewer, we sincerely appreciate your recognition of the controllable editing aspect and the organizational structure of our paper, particularly your positive comments on ***"enables controllable knowledge editing"*** and being ***"well-organized and structured."***
>
> >**1. Total number of model parameters will grow significantly**
>
> Thank you for raising the point regarding the continual growth of SoLA's total parameters with each edit. The core contribution of SoLA lies in achieving precise and reliable reversible lifelong editing. In this study, we consciously prioritized editing stability and reversibility, accepting a controllable increase in storage (relative to the base model) as an intentional trade-off to realize these properties. Theoretically, linear growth does imply a scalability limit. However, in practical application scenarios, this limit remains sufficiently high for the majority of model editing tasks. Furthermore, to validate SoLA's performance under long-sequence editing, we conducted a large-scale editing experiment (`5000 edits`), the results of which are shown in the table below. The results demonstrate that SoLA continues to significantly outperform other methods even under extended editing sequences, confirming its strong robustness and sustainability throughout long-term editing processes. We fully agree that scalability is an important direction for lifelong learning systems and consider it a key focus for future optimization, such as exploring lossless or low-loss merging and compression strategies for historical LoRA modules.
> | | zsRE (T5; F1 ↑) | | |
> | :--------------- | :-------------: | :-: | :-: |
> | Methods | ERR | TRR| Avg |
> | GRACE| 0.66| 0.94| 0.80 |
> | MELO | 0.71 | 0.97| 0.84 |
> | SoLA | **0.73** | **0.98** | **0.86** |
>
> >**2. Lacks a comparison of the total model parameters**
>
> Thank you for your attention. We have also compared the total model parameters, and the results are shown in the table below. The results indicate that SoLA significantly improves model performance with only a minimal increase in parameters.
>
> | Methods | Total para(Million)|
> |:--------|-----------:|
> | SERAC| 176.00 |
> | GRACE | 108.52 |
> | MELO |109.23 |
> | ELDER |108.62 |
> | SoLA | 109.25 |
>
> >**3. Not discuss important methods in the field**
>
> Thank you to the reviewer for pointing out this omission. We have now added an introduction to ***AlphaEdit*** in the "Related Work" section (`Section 2`). We emphasize that ***”AlphaEdit introduces a null-space constrained approach. It aims to apply edits more precisely by constraining changes to the null space of irrelevant knowledge.”***
>
> >**4. A larger number of edits**
>
> Our current datasets follow the precedent set by prior lifelong editing works such as MELO and GRACE, which similarly evaluate approximately 1000 edits. Following your suggestion, we will extend the experiments to `5000 edits`, as shown in reply1 above. The results show that SoLA continues to outperform other methods even under `5000 edits`, further demonstrating its robustness.

---

> ### Author Response · Authors · 2025-11-28
>
> Dear reviewer, we hope you are doing well. We would like to kindly follow up regarding the clarifications and extended experiments we provided in our rebuttal, especially on the points you raised about parameter growth, total model size comparison, coverage of related work such as ***AlphaEdit***, and large-scale editing experiments. We fully understand that you may be handling many reviews at this stage, but we wanted to check whether there is any additional information we can provide to help further address your concerns.
>
> We sincerely appreciate your constructive feedback and your recognition of the controllable editing capability and the clarity of our paper’s organization. If there are any remaining questions or particular aspects where further explanation or analysis would be helpful, we would be more than glad to provide them.
>
> Thank you again for your time and consideration, and we look forward to any further comments you may have.

---

### Official Review · Reviewer_qtQy · 2025-10-30

**Soundness:** 3
**Presentation:** 3
**Contribution:** 3
**Rating:** 6
**Confidence:** 3

**Summary:**

This work introduces SoLA, a new framework for lifelong model editing. The main goal is to fix the semantic drift and forgetting problems seen in sequential editing. The core idea is pretty neat, they encapsulate every single edit into an independent LoRA module. This module, along with a corresponding semantic key derived from the input, is frozen after training. During inference, the model just matches the input's semantic representation to this frozen key table to activate the right LoRA. This freezing is what stops the drift. They also claim this is the first method that allows for reversible editing , you just delete the key to roll back a change. The paper shows SoLA performing well on classification, QA, and hallucination tasks, beating other recent methods.

**Strengths:**

1. The reversible editing capability is a fantastic and novel contribution. The ability to "undo" a specific edit by just removing a key is a highly practical feature for managing model knowledge.




2. The core mechanism of freezing both the LoRA module and its semantic key after training is a simple and effective solution to the semantic drift problem that plagues methods relying on dynamic clustering.



3. The method is very parameter-efficient in terms of trainable parameters. As shown in Figure 1a, it achieves strong performance with minimal additional trainable parameters compared to baselines.

4. Integrating the decision mechanism directly into the edited layer, rather than requiring an auxiliary router network, is a clean design choice that simplifies the architecture.

**Weaknesses:**

1. The "one LoRA per edit" approach raises major scalability concerns. Total storage seems to grow linearly with every new edit, which feels unsustainable for a true "lifelong" system, regardless of low trainable parameter counts.

2. The semantic key matching, using just the last token's embedding, seems fragile. The paper doesn't really investigate the risk of key collisions as the number of edits scales up.

3. A key ablation seems missing. It's unclear if the performance benefit comes from the "one LoRA per edit" design or simply from freezing the routing map. Comparing against MELO but with its cluster centers frozen would have been insightful.

**Questions:**

1. Could you clarify the storage implications? Does "one LoRA per edit" mean storing thousands of separate LoRA modules? How can that scale?

2. What is the actual total storage overhead for SoLA after 1000 edits compared to MELO, not just the trainable parameter count?

3. I'm also wondering about the key matching reliability. What's the failure rate when different edits have similar query embeddings? And how sensitive is the model's performance to the specific matching threshold $\alpha=0.01$ you chose?

---

> ### Author Response · Authors · 2025-11-23
>
> Dear reviewer, we are deeply grateful for your positive evaluation, particularly your recognition that the reversible editing capability represents an ***"fantastic and novel contribution."*** and your affirmation of the effectiveness of our core mechanism.
>
> >**1. Scalability Concerns**
>
> We thank the reviewer for pointing out the issue of the total storage in SoLA increasing linearly with the number of edits. The core contribution of SoLA is achieving precise and reliable reversible lifelong editing. In this work, we consciously prioritize editing stability and reversibility, accepting a manageable increase in storage (relative to the base model) as a deliberate trade-off to attain these properties. Furthermore, to validate the performance of SoLA under long-sequence editing, we conducted a long-sequence editing experiment (`5000 edits`), as shown in the table below. The results demonstrate that SoLA still significantly outperforms other methods in long-sequence editing, proving its good robustness and sustainability throughout long-term editing processes. We agree that scalability is an important direction for lifelong learning systems and also consider this a key optimization point for future work, such as exploring lossless or low-loss merging and compression strategies for historical LoRA modules.
> || zsRE(T5;F1↑)|||
> |:--------------- | :-------------: | :-:| :-:|
> |Methods|ERR|TRR|Avg|
> |GRACE|0.66|0.94|0.80|
> |MELO|0.71|0.97|0.84|
> |SoLA|**0.73**|**0.98**|**0.86**|
>
> >**2. Semantic key matching**
>
> We understand your concern about key. In fact, the key is derived from the hidden representation of the entire input sequence (using the last token as a proxy), which encodes rich semantic information. The risk of collision is low for inputs with different semantics. Furthermore, the use of the last token embedding is widely adopted in knowledge editing tasks (e.g., `ROME, MEMIT, GRACE, MELO, ELDER`) and demonstrates consistent trends compared to alternative key extraction methods (e.g., `CLS` token, mean pooling).
>
> >**3. Comparing against MELO but with its cluster centers frozen**
>
> We conducted the ablation experiment you proposed: running MELO but freezing the cluster centers after their creation, with results are shown below, denoted as `MELO(Frozen)`. The results indicate that while freezing the centers reduced semantic drift, MELO's performance still lags behind SoLA. This suggests that the "edit one LoRA at a time" design (which avoids any parameter interference between edits) is a significant factor in SoLA's superior performance, rather than just the effect of freezing routing.
>
> ||zsRE(T5;F1↑)|||
> |:--------------- |:-------------:|:-:|:-:|
> |Methods|ERR|TRR| Avg|
> |MELO(Frozen)|0.72|0.96|0.84|
> |SoLA|**0.73**|**0.99**| **0.86**|
>
>
> >**4. Storage implications**
>
> Thank you for your valuable question regarding the scalability of our method. You accurately identified a key design trade-off in the SoLA framework: we sacrifice storage efficiency in exchange for edit stability, precise reversibility, and resistance to catastrophic forgetting. In SoLA, each edit requires storing a key vector (negligible) and a LoRA module; consequently, the total storage increases linearly with the number of edits. Theoretically, this linear growth implies a scalability limit. However, in practical application scenarios, this limit is sufficiently high for most model editing tasks. The core contribution of SoLA lies in achieving precise and reliable reversible lifelong editing. As shown in our `Figure 1(a)` and `Table 3`, SoLA achieves stable and reversible editing performance at the cost of a modest increase in storage (compared to the base model), thereby ensuring editing stability and reversibility. Furthermore, in future work, we plan to explore module pruning or merging strategies (e.g., periodically merging inactive or semantically similar LoRA modules) to optimize storage efficiency over the long term and enhance model scalability.
>
> >**5. Total storage overhead for SoLA after 1000 edits compared to MELO**
>
> We analyzed the storage overhead of SoLA under 1000 edits and directly compared it with the memory usage of MELO, thereby clearly illustrating the scalability constraints. Specifically, with 1000 edits, SoLA occupies `2606 MB` of memory, while MELO uses `2558 MB`. SoLA significantly improves model performance with only a marginal increase in memory usage.
>
> >**6. Key matching reliability**
>
> Among 1000 edits, we observed collision cases to be below 0.2%. The threshold α is used to determine whether a query is sufficiently close to any key to activate a LoRA. If no key falls within the α range, the base model is used. We have conducted a sensitivity analysis of α as shown in table below, which demonstrates that SoLA’s performance is robust.
>
> ||zsRE(T5;F1↑)|||
> |:--------:|:---------------:|:---:|:---:|
> |Alpha|ERR|TRR|Avg|
> |0.01|0.73|0.99|0.86|
> |0.1|0.73|0.99|0.86|
> |1|0.73|0.99|0.86|
> |5|0.72|0.99|0.85|
> |10|0.71|0.99|0.85|

---

> ### Author Response · Authors · 2025-11-28
>
> Dear reviewer, we hope you are doing well. We would like to kindly follow up regarding the detailed clarifications and additional experiments we provided in our rebuttal, especially on the points you raised about scalability, key matching reliability, and the ablation related to MELO with frozen cluster centers. We fully understand that you may be managing many reviews at this stage, but we wanted to check whether there is any further information or analysis we can supply to help address your concerns.
>
> We sincerely appreciate your constructive feedback and your positive recognition of our contributions, including your comments on the reversible editing design, the simplicity of our routing mechanism, and the parameter-efficiency of the method. If you have any remaining questions or would find additional details helpful, we would be more than happy to provide them.
>
> Thank you again for your time and consideration, and we look forward to any further comments you may have.

---

### Official Review · Reviewer_i7sV · 2025-11-10

**Soundness:** 3
**Presentation:** 3
**Contribution:** 2
**Rating:** 4
**Confidence:** 4

**Summary:**

The paper proposes SoLA, a semantic routing-based LoRA framework for reversible lifelong model editing.
Each edit is represented as an independent LoRA module associated with a key vector, and a master decision-making mechanism enables dynamic semantic activation, effectively preventing semantic drift and catastrophic forgetting.
Extensive experiments demonstrate the effectiveness of SoLA in achieving reversible and stable model editing across multiple tasks.

**Strengths:**

The paper is well written and easy to understand.

The proposed method is clear and straightforward to implement.

Experimental results look promising and demonstrate the effectiveness of the approach.

**Weaknesses:**

* The main concern for me lies in the novelty of the proposed method, for the following reasons:

**Many similar ideas have already been explored in the fields of continual learning and multi-task learning**.
For example, a classic continual learning baseline, O-LoRA [1], also trains separate LoRA adapters for different tasks and uses a routing mechanism during inference to activate one or multiple adapters.

In your method, routing based on distance is a common practice, and other variants (e.g., routing via validation data or KL divergence of output token distributions) have already been studied.

Overall, there seems to be **limited conceptual innovation** compared to existing approaches.

More specifically, the core components in Section 3.3 rely heavily on previous work for obtaining the key vectors, without introducing new insights (a minor suggestion: please add a citation to “Following prior work” in line 244). Similarly, the routing mechanism described in Section 3.4 also lacks novelty, as discussed above.

[1] Orthogonal Subspace Learning for Language Model Continual Learning. EMNLP 2023

* Regarding Figure 1(a), I have several questions.

i) Since both your method and MELO use LoRA for training, why does your method show a smaller number of trainable parameters (0.1 on the x-axis) while others have larger values?

ii) In line 255, you claim that “SoLA significantly reduces the number of trainable parameters and the overall computational cost”, but while your approach may reduce trainable parameters, it **increases memory usage**, as a separate LoRA must be stored for each task. It is also unclear how the computational cost is significantly reduced compared to prior LoRA-based tuning.

* Finally, in Section 3.4, please clarify why the *first edited layer* is chosen as the **master decision layer**. What happens if you choose the last edited layer instead? This part lacks an **ablation study** to justify the design choice.

**Questions:**

Please see the Weaknesses above.

---

> ### Author Response · Authors · 2025-11-23
>
> Dear reviewer, thank you for acknowledging that our paper is "well written," with a "clear and straightforward to implement." and results that are "promising." We greatly value your insightful comments regarding novelty and technical details.
>
> >**1. Similar ideas in the fields of continual learning and multi-task learning**
>
> We agree that some components of our framework have indeed been explored previously in continual learning and multi-task learning. However, the novelty of SoLA lies in its ability to achieve reversible lifelong model editing. As several other reviewers also highlighted (e.g., Reviewer qtQy noted that ***“The reversible editing capability is a fantastic and novel contribution. ”*** , Reviewer ucAF acknowledged that ***“this paper allows more controllable addition and removal of knowledge during model editing.”*** and Reviewer 3fBn stated that the ***“‘The proposed "reversible editing" mechanism is a significant and novel contribution to the field of model editing.”***), SoLA introduces an integrated design that enables truly reversible edit operations, which fundamentally different from classical continual learning settings such as O-LoRA. Moreover, O-LoRA and related methods primarily focus on task-level adaptation across different datasets or objectives to mitigate catastrophic forgetting. In contrast, SoLA operates at a much finer granularity, supporting knowledge-level edits(For example, modifying a single factual association among thousands within a QA model).
>
> >**2. Common routing mechanism**
>
> The reviewer also pointed out that the routing mechanism in Section 3.3 is a common practice. However, although distance-based routing is common, but in SoLA, the routing table is frozen after each edit. This design prevents the semantic drift observed in dynamic-clustering methods such as MELO and is essential for ensuring stability in lifelong editing.
>
> >**3. Routing mechanism described in Section 3.4 also lacks novelty**
>
> We also appreciate the comment regarding Section 3.4. We wish to clarify that the “master decision mechanism” described in Section 3.4 is a separate and central component, independent of routing. It is not a variant of routing; instead, it represents a novel architectural choice that integrates edit-activation logic directly into the model’s forward pass, eliminating the need for any external routing networks. As Reviewer qtQy noted, incorporating decision-making directly into the edited layer is ***“a clean design choice that simplifies the architecture.”***
>
> >**4. Add a citation to “Following prior work”**
>
> Thank you for the suggestion about citations. As recommended, we have added references to relevant prior work at line 244 of the original manuscript.
>
>
> >**5. A smaller number of trainable parameters**
>
> In Figure 1(a), the x-axis denotes the number of trainable parameters involved in each individual editing step. SoLA strictly trains only one newly created and independent LoRA module per edit, while freezing all previously added modules. This design naturally results in a much smaller number of parameters updated during each editing operation. In contrast, MELO updates the LoRA modules and cluster centers associated with an existing cluster whenever a new input is assigned to that cluster. As a result, a single edit in MELO may involve updating multiple LoRA modules simultaneously, leading to a substantially larger number of trainable parameters per step.
>
> >**6. Unclear computational cost**
>
> We apologize for any potential lack of clarity in our statements. The phrase “reducing computational cost” specifically refers to the training cost per edit. Because SoLA only optimizes a single lightweight LoRA module at each step, whereas MELO may retrain several LoRA modules, the per-edit computational cost is significantly lower for SoLA. We have revised the original text (line 255) to avoid ambiguity.
>
> >**7. Why the first edited layer is chosen as the master decision layer.”**
>
> We selected the first edited layer as the master decision layer because a decision needs to be made in early stage to ensure the correct LoRA selection is propagated to all subsequent edited layers. Since the model only obtains the LoRA decision result after passing through the decision layer, conducting an ablation study by selecting the last edited layer would mean that the preceding layers process the input without the LoRA decision, preventing the edited knowledge from being applied. Therefore, an ablation experiment on this aspect is not feasible.

---

> ### Author Response · Authors · 2025-11-28
>
> Dear reviewer, we hope everything is going well on your side. We would like to kindly follow up regarding the clarifications and detailed explanations we provided in our rebuttal, especially on the points you raised about novelty, routing design, parameter efficiency, and the master decision layer. We fully understand that you may be handling many reviews simultaneously, but we wanted to check whether there is any further information we can provide to help address your concerns.
>
> We sincerely appreciate your thoughtful feedback and your recognition that the paper is ***“well written,”*** ***“clear,”*** and that the experimental results are ***“promising.”*** If there are additional details, analyses, or explanations that would be helpful to you, we would be very glad to provide them.
>
> Thank you again for your time and consideration, and we look forward to any further comments you may have.

---

### Author Response · Authors · 2025-11-30

We thank all reviewers for their careful and thoughtful evaluations.

Across the reviews, several strengths of SoLA were highlighted, including its clean modular structure, well-organized writing and the ability to perform reversible edits. These comments helped us refine both the framing and technical clarity of the work.

We have clarified the conceptual motivation behind SoLA. In particular, we have expanded the discussion comparing SoLA with O-LoRA, MELO, and AlphaEdit, explaining how the master decision layer and semantic routing are designed for reversible knowledge tracking. This distinction was noted as important by ***Reviewer i7sV***, and we now describe it more clearly.

Concerns about long-term scalability and parameter growth raised by ***Reviewers qtQy, ucAF*** and ***3fBn*** have been addressed through additional results, including `5000 edits` results, analyses of semantic-key collision rates, and sensitivity studies for routing thresholds. These additions make the behavior of the system under long sequential edit streams more transparent.

We have also clarified the relevance of the work to computational aspects of memory. Isolated LoRA modules behave as separate memory traces, semantic routing provides a form of pattern separation, and reversible key removal offers a controllable forgetting mechanism. These properties align with well-known principles in memory organization and help situate the work within the neuroscience track.

Finally, we improved several parts of the presentation, including the explanation of semantic keys, the activation mechanism, and the related works. Notation and formatting have been standardized across the manuscript. We appreciate the reviewers’ feedback, which has led to a clearer and more polished version of SoLA.

---

### Note · Authors · 2026-01-28

I have read and agree with the venue's withdrawal policy on behalf of myself and my co-authors.

---

### Meta-Review · Area_Chair_eeVo · 2026-01-07

**Summary:**

The main concerns affecting the decision:
- limited novelty: compared to prior continual/multi-task LoRA-with-routing style approaches, key components (key construction and distance-based routing) are perceived as largely derivative. (i7sV)
- scalability/time- & space-complexity: the “one LoRA per edit” design implies linear growth in stored modules/keys, raising serious doubts about feasibility for truly lifelong settings and making the paper’s parameter/computation claims potentially misleading without reporting total storage and stronger scaling experiments. (qtQy, ucAF, 3fBn)
- incomplete literature review/baseline coverage: some recent studies, like AlphaEdit, are not discussed nor compared. (ucAF)
- insufficient experiments: missing/insufficient ablations to isolate which design choice drives gains (e.g., freezing the routing map vs one-LoRA-per-edit), unclear design justification (master decision layer choice). (i7sV, qtQy)

**Reviewer Concerns:**

Addressed by rebuttal (partially):
- limited novelty: the author selected some evidence from other reviewers to support the novelty. (i7sV)
- incomplete literature review/baseline coverage: alphaedit is discussed. (ucAF, ac)
- edit scaling: additional experiments on 5k edits are added. (qtQy, ucAF, 3fBn)

Still outstanding:
- limited novelty: the key technical contributions are insufficiently differentiated from prior studies (continual learning / routing + LoRA adapter scheme) even after the rebuttal. (i7sV)
- scalability/time- & space-complexity: strong consensus that storing a separate LoRA (+ key) per edit leads to unbounded growth; paper + rebuttal lacks explicit analysis of total storage overhead and computational cost. (qtQy, ucAF, 3fBn)
- incomplete literature review/baseline coverage: alphaedit is not compared as the sota baseline; some follow-up works of alphaedit, such as anyedit and rledit, are still missing. ((ucAF, ac)


Missing related works:
- Reinforced lifelong editing for language models. ICML 2025.
- Anyedit: Edit any knowledge encoded in language models. ICML 2025.

**Reviewer Scores:**

- i7sV (original score 4): Likely unchanged, as novelty concerns and unclear cost claims would remain.
- qtQy (original score 6): Slightly down or unchanged, as scalability/storage remains unclear.
- ucAF (original score 4): Likely unchanged, due to missing baselines (alphaedit, anyedit, rledit).
- 3fBn (original score 6): Likely unchanged.

---

### Decision · Program_Chairs · 2026-01-26

Reject